## Article

# Transformation of Light Alkanes into High-Value Aromatics

**Muhammad Naseem Akhtar**

Interdisciplinary Center for Refining and Advanced Chemicals (CRAC), King Fahd University of Petroleum & Minerals, Dhahran 31261, Saudi Arabia; mnakhtar@kfupm.edu.sa; Tel.: +966-568518462

**Abstract:** This research work is focused on the transformation of light alkane (propane) into high-value aromatics using gallo-alumino-silicate catalysts. Two sets of gallo-alumino-silicates were synthesized for this study. In the first set, the ratio of Ga/(Al+Ga) was modified, while the Si/(Al+Ga) ratio was held constant. In the subsequent set, the Si/(Al+Ga) ratio was adjusted, while maintaining a consistent Ga/(Al+Ga) ratio. This approach aimed to directly assess the impact of each ratio on catalyst performance. The comprehensive characterization of all catalysts was conducted using various instrumental techniques, i.e., BET surface area, XRD, $NH_3$-TPD, $^{27}Al$, $^{71}Ga$ and $^{29}Si$ MAS NMR, and XPS. A gradual reduction in the percentage of crystallinity and rise in meso-surface area was noticed with a rise in Ga/(Al+Ga) ratio. The total acidity ($NH_3$-TPD) demonstrated a decline as the Si/(Al+Ga) ratio increased, attributed to an overall decline in $Al^{3+}$ or $Ga^{3+}$ species. The XPS intensity of the Ga $2p_{3/2}$ peak rose in correlation with an elevated ratio of Ga/(Al+Ga), suggesting the formation of extra-framework Ga species. The propane conversion, aromatic yield, and aromatization/cracking ratio exhibited an increase with an increasing Ga/(Al+Ga) ratio, reaching an optimum value of 0.46 before declining. Conversely, an appreciable drop in the conversion of propane and yield of aromatics was detected with the rise in Si/(Al+Ga) ratio, attributing to the decline in acidity. The catalyst having a Ga/(Al+Ga) ration of 0.46 exhibited the highest propane conversion and aromatic yield of 83.0% and 55.0% respectively.

**Keywords:** propane aromatization; GaAlMFI; zeolite; Ga/Al ratio; Si/(Al+Ga) ratio; calcination temperature; acidity

## 1. Introduction

The aromatization of light alkanes is a significant process from academic as well as industrial points of view. Indeed, the discovery of shale gas in the 21st century has stimulated the conversion of light gases into higher value aromatics (BTX). Indeed, BTX are essential intermediates in the chemical industry. They are employed in the synthesis of a broad spectrum of chemicals and materials, e.g., various polymers, plastics, resins, solvents, dyes, and pharmaceuticals, etc. The prospect of producing BTX directly from light gas resources holds great potential as it reduces the heavy reliance on crude oil resources. The dehydro-aromatization of light gases generally occurs over bifunctional catalysts through a complicated network of intermediate reactions.

The gallium-containing catalysts are most widely acknowledged for propane aromatization [1–5]. Although the precise mechanism is still not clear, the prevailing proposed pathway indicates a stepwise reaction facilitated by catalysts possessing dual functionalities. In this process, propane is activated and cracked to $C_1$ and $C_2$ species at Brønsted acid sites of the zeolite and dehydrogenated over Ga species, resulting in the formation of olefinic intermediates. These intermediates then further undergo several different reactions leading to the formation of aromatic products [2,6–8]. The extent of this interaction becomes a key factor determining the overall performance of the catalyst. A detailed mechanism for the aromatization of light alkanes ($CH_4$, $C_2H_8$, $C_3H_{10}$, $C_4H_{12}$, etc.) has been reviewed [9] along with different factors affecting the aromatization. The metal-containing ZSM-5 catalysts exhibit higher selectivity to aromatics. The metal species facilitates the dehydrogenation of

alkanes to the respective alkenes due to higher Lewis acidity [10–14]. Studies have indicated that MFI zeolites containing gallium can be regarded as the most effective catalysts for aromatization as compared with other transition metals. The conversion of propane to aromatics has been reported across a range of gallium-based MFI zeolites, including the mechanical mixing of gallium oxide and ZSM-5, gallium-impregnated ZSM5 [15–20], gallo-silicates [21–23], and gallo-alumino-silicates [24–26]. The detection of different gallium species is pivotal in formulating suitable catalysts, as the performance of a catalyst is significantly affected by the gallium's specific state. In one study [1], gallo-alumino-silicates were treated with HCl to remove Ga species that have feeble interactions with the zeolite. This treatment led to the preservation of the most favorable species for propane activation and resulted in an improvement in propane aromatization performance.

It is important to mention that several studies have been reported to perform the aromatization of light alkanes by the addition of gallium metal to the ZSM-5. This was achieved following different methods, e.g., incipient impregnation, wet impregnation, ion-exchange, or hydrothermal synthesis. The primary goal of this study was to investigate the precise influence of the ratios of Ga/(Al+Ga) and Si/(Al+Ga) on the performance of gallo-alumino-silicates during the conversion of propane to aromatics.

## 2. Results and Discussion

This study involved the preparation of two series of gallo-alumino-silicate catalysts. In the initial series, changes were implemented in the Ga/(Al+Ga) ratio while keeping the Si/(Al+Ga) ratio constant. Conversely, in the second series, the ratio of Si/(Al+Ga) was adjusted while maintaining a constant Ga/(Al+Ga). Further information about these zeolites can be found in Table 1.

**Table 1.** Metal analysis (mass%) and ratio of different metals in the catalyst samples based on the ICP analysis.

| Catalyst Code | Mass% | | | Molar Ratio | |
|---|---|---|---|---|---|
| | **Al (%)** | **Si (%)** | **Ga (%)** | **Si/(Ga+Al)** | **Ga/(Ga+Al)** |
| Cat-1 | 1.20 | 22.24 | 0.00 | 17.79 | 0.00 |
| Cat-2 | 1.14 | 22.19 | 0.48 | 16.06 | 0.14 |
| Cat-3 | 1.22 | 23.13 | 1.00 | 13.80 | 0.24 |
| Cat-4 | 1.09 | 23.78 | 1.15 | 14.88 | 0.29 |
| Cat-5 | 0.65 | 22.19 | 1.43 | 17.69 | 0.46 |
| Cat-6 | 0.47 | 22.85 | 1.61 | 20.12 | 0.57 |
| Cat-7 | 1.54 | 23.48 | 1.55 | 10.53 | 0.28 |
| Cat-8 | 0.87 | 34.21 | 0.97 | 26.32 | 0.30 |
| Cat-9 | 0.90 | 58.22 | 0.99 | 43.58 | 0.30 |
| Cat-10 | 0.84 | 81.03 | 0.93 | 65.00 | 0.30 |

### 2.1. Characterization of Catalysts

Both series of zeolites were characterized using various instrumental techniques and results are discussed as below:

All catalyst samples were analyzed by ICP-OES to identify the percentage of metals (Ga, Al, and Si). The findings are presented in Table 1. In the first series of catalyst samples (Cat-1 to Cat-6), the Ga/(Al+Ga) molar ratio was changed from 0.0 to 0.6. Whereas, in the second series of catalyst samples (Cat-7 to Cat-10), the Si/(Al+Ga) molar ratio was varied from 11 to 65.

The XRD patterns of zeolites and their relative percent crystallinity are illustrated in Figure S1 and Table 2, correspondingly. The XRD patterns of all the catalyst samples indicated the existence of crystalline MFI samples. The XRD reflections of $Ga_2O_3$ particles

were not observed [27] meaning that all Ga species were present in the framework of MFI. It was observed that the relative crystallinity was gradually dropped from Cat-1 to Cat-6 with increase in the Ga/(Al+Ga) ratio. This drop can be correlated to the bigger size of the $Ga^{3+}$ species (0.62 Å) as compared with that of the $Al^{3+}$ species (0.51 Å). The increase in concentration of the $Ga^{3+}$ species causes a stress on the crystal lattice resulting in the drop of crystallinity. There was no difference in relative crystallinity with the change in the Si/(Al+Ga) ratio.

**Table 2.** Structural and textural properties of catalyst samples.

| Catalyst Code | Crystallinity (XRD) | Textural Properties | | | | |
|---|---|---|---|---|---|---|
| | | $S_{BET}$ $(m^2 g^{-1})$ | Micro Pore Area $(m^2 g^{-1})$ | External Area $(m^2 g^{-1})$ | $V_{Total}$ $(cm^3 g^{-1})$ | $V_{Micro}$ $(cm^3 g^{-1})$ |
| Cat-1 | 90 | 285.0 | 138.0 | 147.0 | 0.560 | 0.060 |
| Cat-2 | 86 | 281.6 | 133.0 | 148.7 | 0.580 | 0.070 |
| Cat-3 | 84 | 283.0 | 134.2 | 148.8 | 0.692 | 0.063 |
| Cat-4 | 83 | 281.8 | 132.3 | 148.9 | 0.768 | 0.070 |
| Cat-5 | 80 | 290.0 | 126.6 | 163.5 | 0.792 | 0.068 |
| Cat-6 | 76 | 289.4 | 121.8 | 168.1 | 0.761 | 0.066 |
| Cat-7 | 87 | 289.0 | 135.0 | 154.0 | 0.570 | 0.060 |
| Cat-8 | 89 | 292.0 | 136.0 | 156.0 | 0.56 | 0.059 |
| Cat-9 | 85 | 280.0 | 138.0 | 142.0 | 0.580 | 0.062 |
| Cat-10 | 88 | 284.0 | 133.0 | 151.0 | 0.530 | 0.061 |

The textural characteristics of zeolites are provided in Table 2. The isotherms of $N_2$ desorption and adsorption exhibited a type-I isotherm indicating that all catalyst samples have a microporous structure. There was no substantial difference in BET surface area and micropore volume with the increase in the Ga/(Al+Ga) ratio, indicating that the addition of $Ga^{3+}$ species does not affect the microporous structure of catalyst samples. However, it was noticed that the mesoporous surface area and the mesopore volume were successively augmented with a rise in the Ga/(Al+Ga) ratio, indicating an improvement in the diffusion of the molecules of reactants and products. However, no significant changes in the mesoporous surface area and the mesopore volume were noticed with the rise in the Si/(Al+Ga) ratio. This observation is further supported in the literature [28] as increases in $S_{meso}/S_{BET}$ was noticed following the isomorphous replacement of Al with Ga.

This acidity of catalyst samples was measured using $NH_3$-TPD and FTIR-pyridine chemisorption techniques. The $NH_3$-TPD profiles for catalyst samples are shown in Figure S2. Two distinct desorption peaks were observed with maxima at 210–240 °C and 400–430 °C, corresponding to the release of $NH_3$ from weak and strong acid sites. The $NH_3$-TPD analysis data for all the catalysts are presented in Table 3 and plotted against changes in Ga/(Al+Ga) and Si/(Al+Ga) ratios in Figures 1 and 2, respectively. The total acidity displayed an opposite trend with respect to both of these ratios. It increased with the increase in the Ga/(Al+Ga) ratio while it decreased in the Si/(Al+Ga) ratio. The FTIR spectra of the chemisorbed pyridine for some selected catalyst samples are depicted in Figure S3, and the data related to the Bronsted and Lewis acid sites are given in Table 3. The peaks around 1545 and 1450 $cm^{-1}$ were used to quantify the Bronsted and Lewis acid sites, respectively. The Bronsted acid sites are generated when –OH groups are attached to the tetrahedral $Ga^{3+}$ or $Al^{3+}$ species present in the framework of zeolite, i.e., Zeolite-Al/Ga-O-H-Py$^{\delta+}$ species (when pyridine is adsorbed on O-H). The extra framework metal ions (not part of framework) are identified as Lewis acid sites (when pyridine is adsorbed on these metal species directly), i.e., Zeolite-Al/Ga-Py$^{\delta+}$ species. It was identified that the concentration of the Bronsted acid sites slightly decreased, whereas the Lewis acid sites

increased with increasing gallium content in the gallo-alumino-silcates. The larger radius of $Ga^{3+}$ relative to $Al^{3+}$ results in a stress on the crystal lattice during isomorphous replacement, leading to the degallation and an increase in the percentage of extra-framework $Ga^{3+}$ species [19]. Therefore, a decrease in $Ga_{Td}$ framework species and an increase in $Ga_{Oh}$ species are responsible for the decrease in the Bronsted acid sites and the increase in the Lewis acid sites.

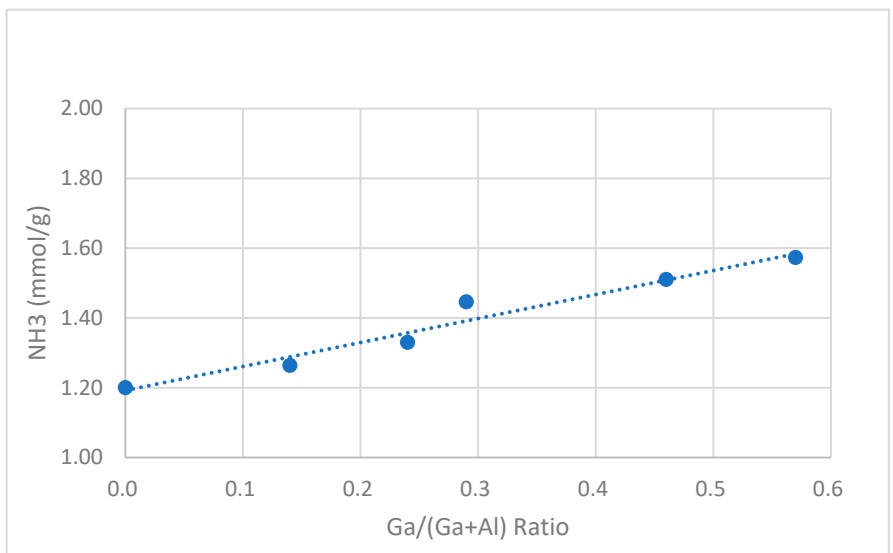

**Figure 1.** Total acidity ($NH_3$-TPD) of catalyst samples against [Ga/(Ga+Al)] ratio.

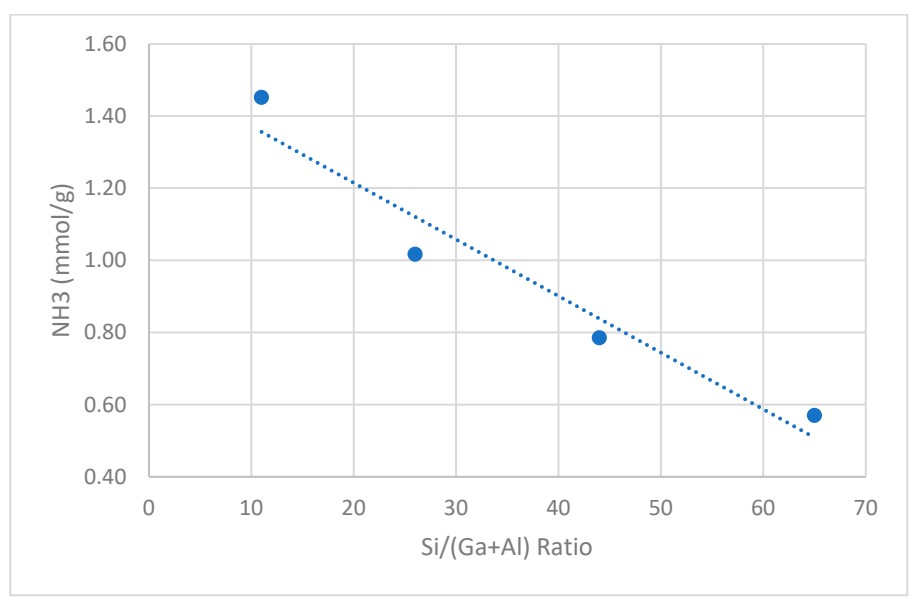

**Figure 2.** Total acidity ($NH_3$-TPD) of catalyst samples against [Si/(Ga+Al)] ratio.

**Table 3.** Acidity of catalyst samples based on $NH_3$-TPD and FTIR-Pyridine adsorption.

| Catalyst Code | Relative Crystallinity (XRD) | $NH_3$ TPD Results (mmol $NH_3$ $g^{-1}$) | | | Acidity Based on Pyridine Adsorbed FTIR (mmol Pyr/g) | | |
|---|---|---|---|---|---|---|---|
| | | LTP | HTP | Total | Bronsted | Lewis | Total (B+L) |
| Cat-1 | 90 | 0.70 | 0.41 | 1.20 | 1.46 | 0.15 | 1.61 |
| Cat-2 | 86 | 0.78 | 0.49 | 1.26 | 1.41 | 0.19 | 1.60 |

**Table 3.** *Cont.*

| Catalyst Code | Relative Crystallinity (XRD) | NH₃ TPD Results (mmol NH₃ g⁻¹) | | | Acidity Based on Pyridine Adsorbed FTIR (mmol Pyr/g) | | |
|---|---|---|---|---|---|---|---|
| | | LTP | HTP | Total | Bronsted | Lewis | Total (B+L) |
| Cat-3 | 84 | 0.85 | 0.48 | 1.33 | 1.37 | 0.26 | 1.63 |
| Cat-4 | 83 | 0.94 | 0.51 | 1.45 | 1.33 | 0.32 | 1.65 |
| Cat-5 | 80 | 1.02 | 0.49 | 1.51 | 1.30 | 0.38 | 1.68 |
| Cat-6 | 76 | 1.05 | 0.52 | 1.57 | 1.26 | 0.45 | 1.71 |
| Cat-7 | 87 | 0.94 | 0.51 | 1.45 | | | |
| Cat-8 | 89 | 0.56 | 0.45 | 1.02 | | | |
| Cat-9 | 85 | 0.40 | 0.38 | 0.78 | | | |
| Cat-10 | 88 | 0.26 | 0.31 | 0.57 | | | |

The $^{27}$Al MAS NMR spectra of the catalyst samples are depicted in Figure 3. All samples displayed two resonances at 50.2 and $-0.2$ ppm, corresponding to the tetrahedral framework ($Al_{Td}$) and octahedral extra framework ($Al_{Oh}$) species, respectively [29]. Overall, no significant differences in the $^{27}$Al NMR of all catalyst samples were observed. However, a slight decrease in in the intensity of the peak (50.2 ppm) related to the $Al_{Td}$ species was observed with an increase in the Ga/(Al+Ga) ratio indicating a slight decrease in the framework Al species. The peak intensity of the $Al_{Td}$ species is very high compared with that of the $Al_{Oh}$ species. Therefore, a slight change in concentration of the $Al_{Td}$ species is more prominent compared with that of the $Al_{Oh}$ species.

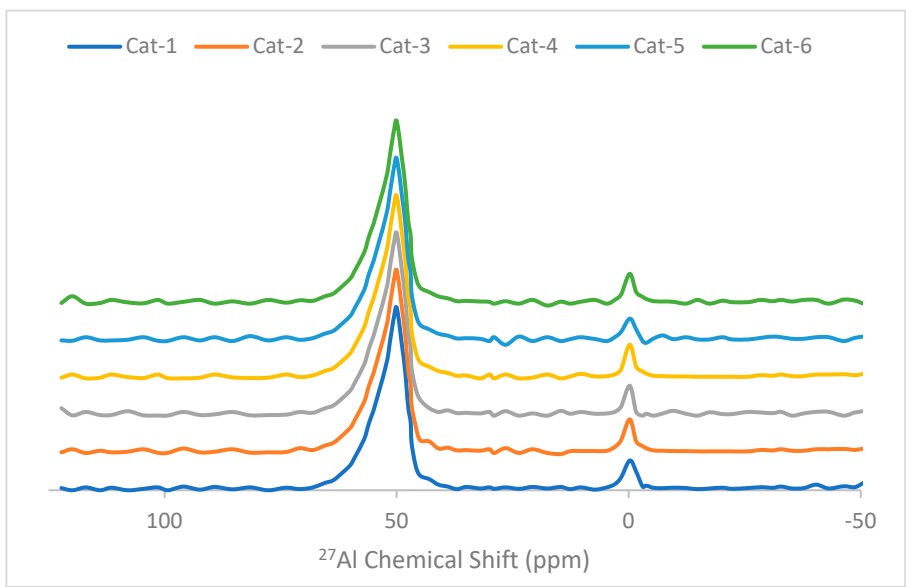

**Figure 3.** $^{27}$Al MAS NMR spectra of catalyst samples.

The $^{71}$Ga MAS NMR spectra of the catalyst samples are presented in Figure 4. All the catalyst samples (Cat-2 to Cat-6) except Cat-1 displayed a single broad peak with a maxima in the range of 156–157 ppm. Since Cat-1 did not contain any gallium, no signal was observed. It has been reported [30] that gallium exists in two different forms in gallo-alumino-silicates. The first $Ga^{3+}$ tetrahedral framework ($Ga_{Td}$) species exhibit a peak at 156 ppm while the second $G^{3+}$ octahedral extra-framework ($Ga_{Oh}$) species exhibit a peak at 50 ppm. However, in most cases, the peak related to $Ga_{Oh}$ was not resolved because of a strong quadrupolar effect [30,31]. The intensity of the peak (156–157 ppm) was

increased with an increase in the Ga/(Al+Ga) ratio, indicating an increase in the tetrahedral framework (Ga$_{Td}$) species.

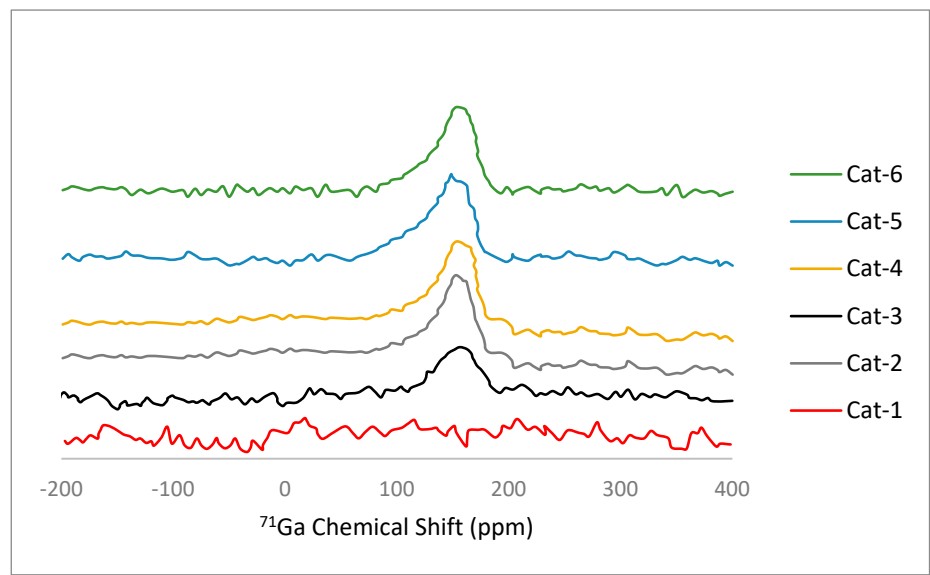

**Figure 4.** $^{71}$Ga MAS NMR spectra of catalyst samples.

The $^{29}$Si MAS NMR spectra of the catalyst samples are depicted in Figure 5. All the catalyst samples exhibited a similar pattern, i.e., a strong peak in the range of −110 to −120 ppm with a small shoulder peak at −101–110 ppm. The deconvolution of these peaks resulted in three peaks, i.e., two strong peaks at −113 and −116 ppm and one small peak at −105 ppm. The two dominating resonances at −113 ppm and −116 ppm were assigned [32,33] to symmetric and asymmetric Si species bonded to four -O-Si groups [i.e., Si-(O-Si)$_4$] in the lattice, respectively. The resonance at −105 ppm is broad with a small intensity and is assigned to Si species bonded to three –O-Si groups along with one O-Al or O-Ga species [i.e., Si-(O-Si)$_3$(O-Ga or O-Al)] [28,31]. It was an ill-defined broad shoulder with maxima at −105.9 in Cat-1, which was further shifted to 104 ppm from Cat-2 to Cat-6. It has been reported [34,35] that the replacement of Al$^{3+}$ by Ga$^{3+}$ in the framework of MFI causes reduction in the chemical shielding by 1–3 ppm.

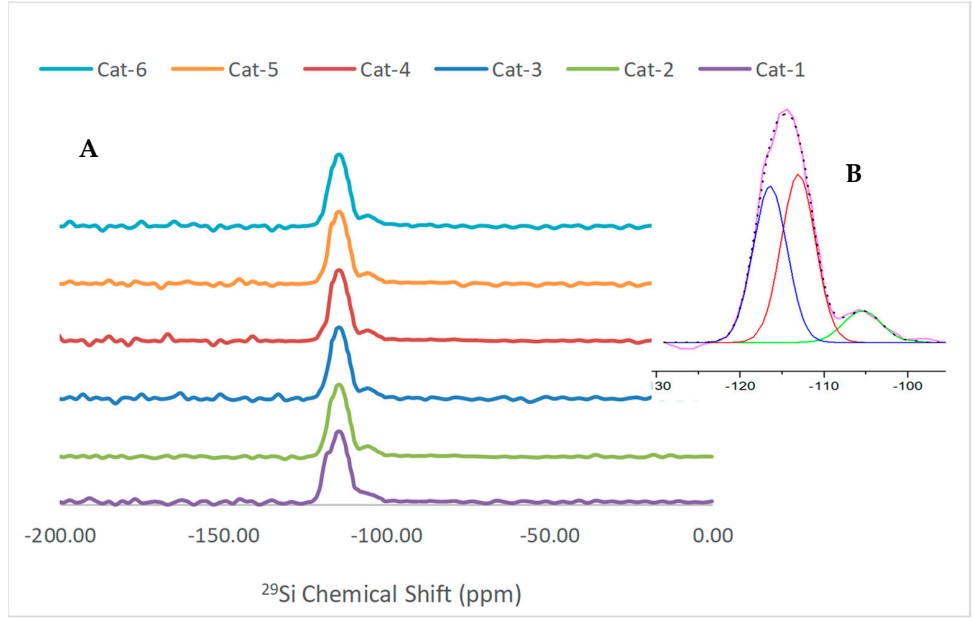

**Figure 5.** $^{29}$Si MAS NMR spectra of catalyst samples (**A**), and deconvolution of peaks (**B**).

The X-ray photoelectron spectra (XPS) of the Ga $2p_{3/2}$ region for Cat-2 to Cat-6 samples are presented in Figure 6. The XPS study of Cat-2 was performed before calcination and after calcination, whereas the XPS study of all the other catalysts was performed after calcination at 550 °C. The Cat-2 (without calcination) sample exhibited a very small peak at 1118.5 eV; however, after calcination, this peak was increased in intensity by several folds. All other catalyst samples exhibited a strong peak with maxima at 1118.5 eV. The intensity of this peak was increased with a surge in Ga/(Al+Ga). It has been reported [36] that the GaAlMFI sample showed a small peak before calcination. However, following calcination, the intensity of this peak was increased by 10 times, indicating the conversion of the framework Ga species to extra-framework Ga species. Therefore, it can be concluded that an increase in the Ga/(Al+Ga) ratio causes more increase in the extra-framework Ga species after calcination.

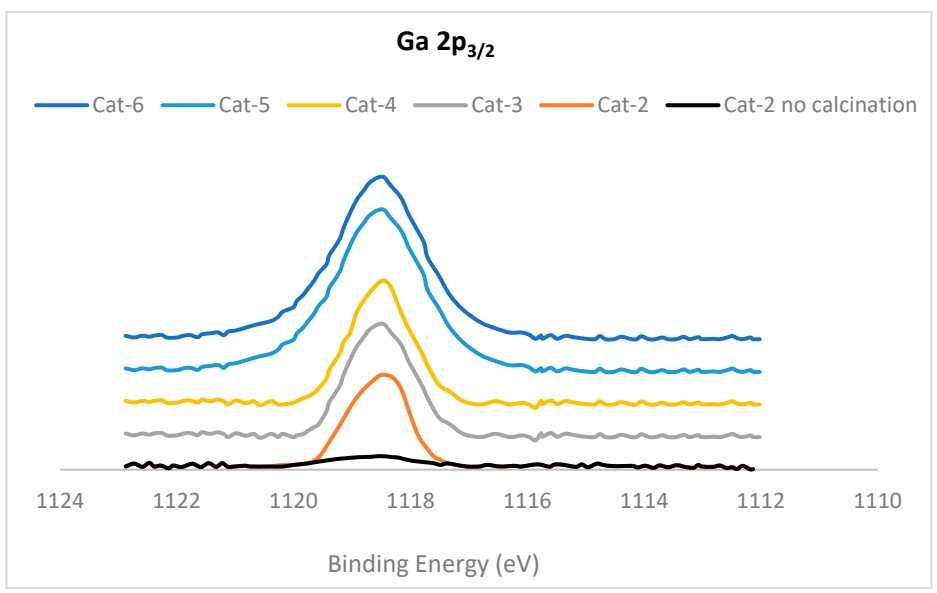

**Figure 6.** X-ray photoelectron spectra (XPS) of Ga $2p_{3/2}$ region of catalyst samples.

The SEM images of the two catalyst samples (Cat-2 and Cat-5), along with the quantitative EDX results, are shown in Figures 7 and 8. Both samples displayed similar morphology with spherical particles having an approximately 10-micron diameter and a uniform distribution of all three elements (i.e., Si, Al, and Ga) among the particles. The elemental mapping of catalyst-2 displayed lower amounts of dispersed Ga compared with that of catalyst-5, because catalyst-2 has a Ga/Al molar ratio of 0.16, compared with that of the Cat-5 sample, which has a Ga/Al molar ratio of 0.85.

## 2.2. Evaluation of Catalysts

The aromatization of propane was performed in a tubular reactor (fixed bed) at 1.0 bar pressure and 550 °C temperature. The conversion and product selectivity were calculated using following equations:

$$\text{Propane conversion (\%)} = \frac{\text{Propane in feed} - \text{Ppropane in products}}{\text{Propane in feed}} \times 100$$

$$\text{Selectivity of product (\%)} = \frac{\text{Yield of product}}{\text{Propane conversion}} \times 100$$

$$\text{Aromatization/Cracking Ratio} = \frac{\text{Selectivity for aromatics}}{\text{Selectivity for C}_1, \text{C}_2, \text{ and C}_2^=}$$

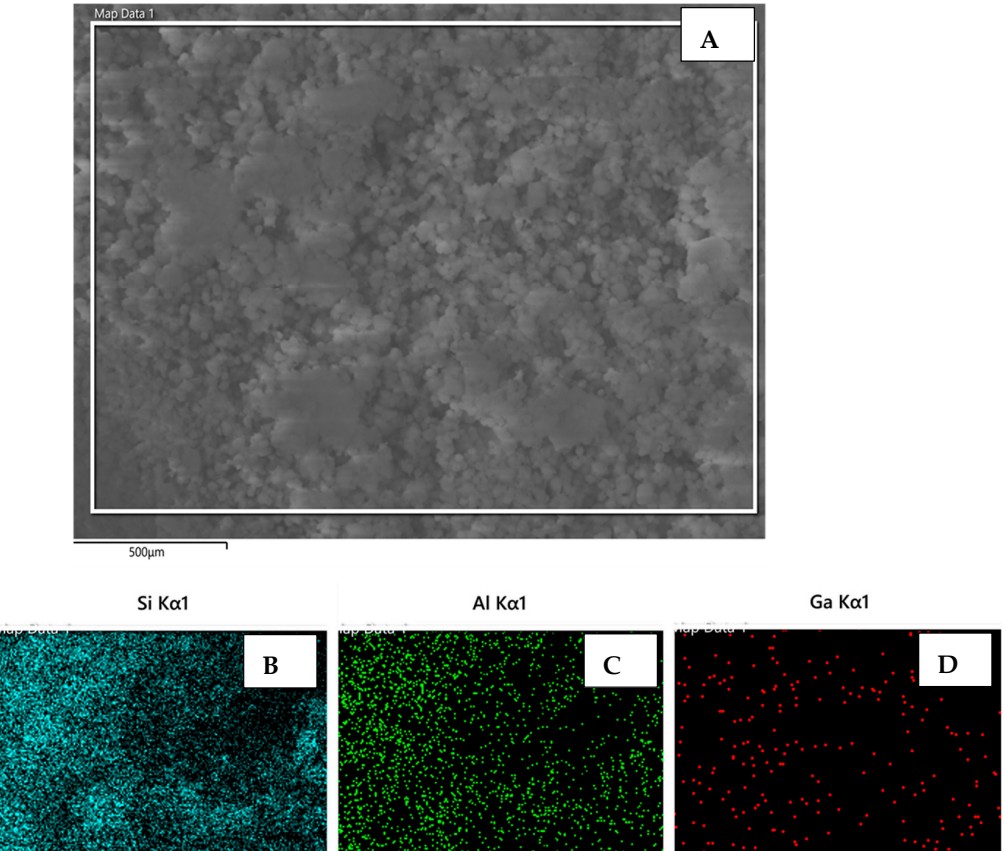

**Figure 7.** (**A**) Scanning electron microscope (SEM) micrograph of catalyst (Cat-2) sample, and elemental mapping of Si, Al, and Ga elements (**B**–**D**), respectively, based on the EDX analysis.

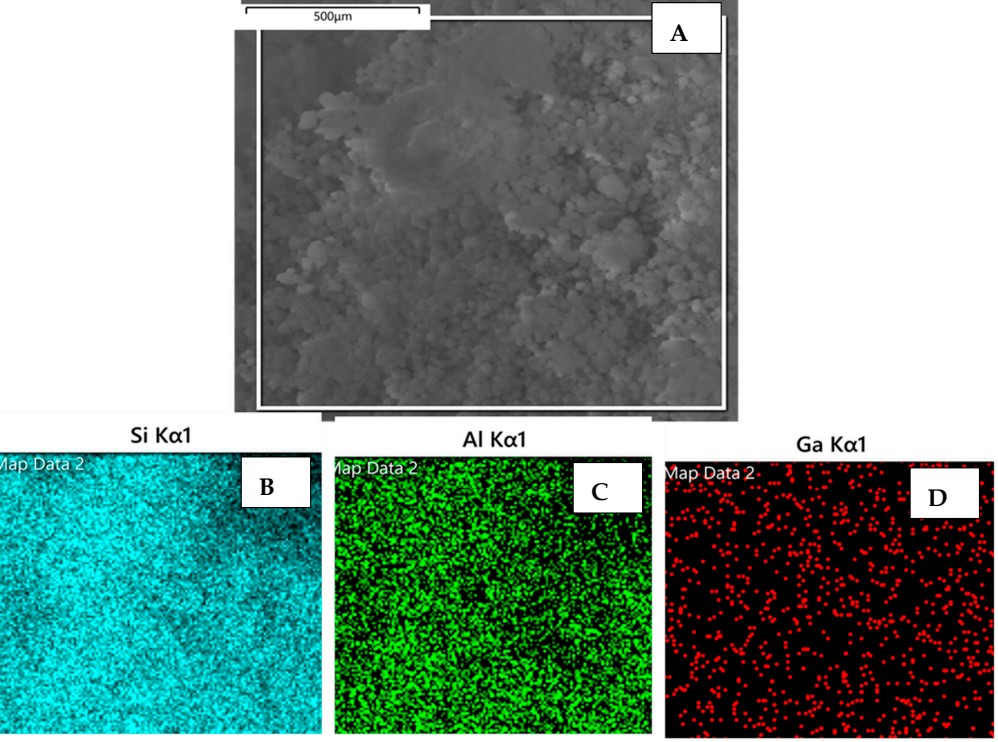

**Figure 8.** (**A**) Scanning electron microscope (SEM) micrograph of catalyst (Cat-5) sample, and elemental mapping of Si, Al, and Ga elements (**B**–**D**), respectively, based on the EDX analysis.

### 2.2.1. Impact of Ga/(Al+Ga) Ratio

In this investigation, a set of gallo-alumino-silicates (Cat-1 to Cat-6) were synthesized by systematically altering the ratio of Ga/(Al+Ga) from 0.0 to 0.6 while maintaining a constant ratio of Si/(Al+Ga). This deliberate choice in the synthesis design enabled the focused examination of the direct impact of the ratio of Ga/(Al+Ga) on the conversion of propane to aromatics. The data related to a catalytic evaluation of this series of catalysts are presented in Table 4 and Figures 9 and 10. The introduction of gallium to the MFI zeolite resulted in an elevation of conversion of propane and aromatic yield.

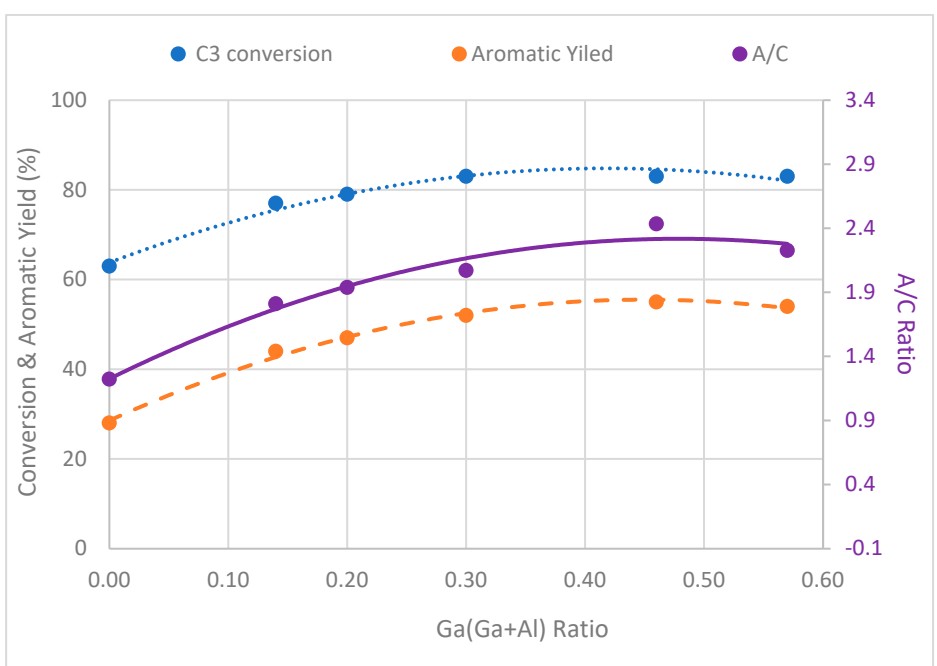

**Figure 9.** Plot of C$_3$ conversion, aromatic yield, and aromatization/cracking (A/C) ratio against [Ga/(Ga+Al)] ratio of catalyst samples.

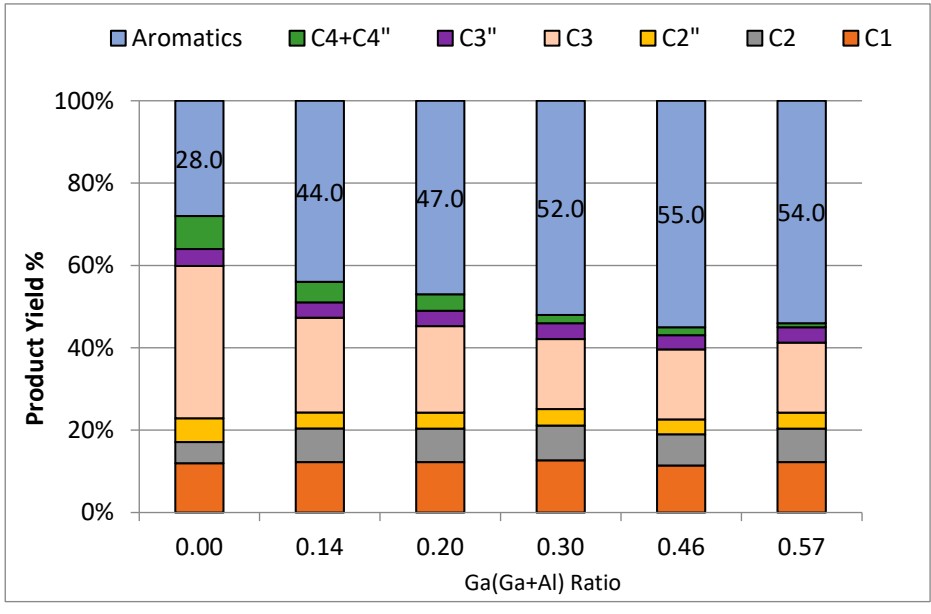

**Figure 10.** Distribution of product against [Ga/(Ga+Al)] ratio of catalyst samples.

**Table 4.** Product distribution with respect to Ga/(Ga+Al) ratio.

| Catalyst Code | Cat-1 | Cat-2 | Cat-3 | Cat-4 | Cat-5 | Cat-6 |
|---|---|---|---|---|---|---|
| Ratio Ga/(Ga+Al) | 0.00 | 0.14 | 0.20 | 0.30 | 0.46 | 0.57 |
| $C_3$ Conversion | 63.0 | 77.0 | 79.0 | 83.0 | 83.0 | 83.0 |
| $C_1$ | 12.0 | 12.2 | 12.2 | 12.7 | 11.4 | 12.2 |
| $C_2$ | 5.2 | 8.2 | 8.1 | 8.4 | 7.6 | 8.1 |
| $C_2^=$ | 5.8 | 3.9 | 3.9 | 4.0 | 3.6 | 3.9 |
| $C_3$ | 37.0 | 23.0 | 21.0 | 17.0 | 17.0 | 17.0 |
| $C_3^=$ | 4.1 | 3.7 | 3.7 | 3.9 | 3.5 | 3.7 |
| $C_4 + C_4^=$ | 8.0 | 5.0 | 4.0 | 2.0 | 1.9 | 1.0 |
| Total Aromatics | 28.0 | 44.0 | 47.0 | 52.0 | 55.0 | 54.0 |

Specifically, propane conversion showed an upward trend, rising from 63% to 83% as the ratio of Ga/(Al+Ga) was elevated from 0 to 0.3. Subsequently, the conversion stabilized, and no further increase was observed with a subsequent increase in the ratio of Ga/(Al+Ga) from 0.3 to 0.57. Similarly, the aromatic yield exhibited a rise from 28% to 55% as the Ga/(Al+Ga) ratio increased from 0 to 0.46. However, a subsequent drop in aromatic yield to 54% was noted with a further increase in the Ga/(Al+Ga) ratio to 0.57. These findings suggest a nuanced relationship between the ratio of Ga/(Al+Ga) and the conversion of propane and aromatic yield, with optimal values observed within specific ranges of the Ga/(Al+Ga) ratio. Therefore, it can be concluded that the Ga/(Al+Ga) ratio of 0.46 is optimum for obtaining the highest propane conversion and aromatic selectivity. This conclusion is further supported by the plot of the aromatization/cracking (A/C) ratio (Figure 9) against the Ga/(Al+Ga) ratio. The A/C displayed its highest value of 2.4 for a Ga/(Al+Ga) ratio of 0.46. It has been well established [20,23,37,38] that the addition of gallium to MFI zeolite improves the conversion and aromatic selectivity. The metal-modified MFI catalysts exhibit bifunctionality due to the existence of Brønsted acid sites attributed to the zeolite structure and Lewis acid sites from metals having inadequate electrons. The aromatization of propane involves a series of reactions at both acid sites within the catalysts [39]. The Brønsted acid sites are typically associated with cracking, oligomerization, isomerization, and cyclization reactions, while the Lewis acid sites play a role in dehydrogenation reactions. Intermediate products shuttle between these acid sites to complete the intricate steps of aromatization. A general reaction mechanism based on the involvement of Bronsted and Lewis acid sites is given in reaction Scheme 1. The superior performance of gallo-alumino-silicate catalysts can be ascribed to the substantial presence of Brønsted acid sites, complemented by Lewis acid sites in the configuration of well-dispersed and reducible extra-framework $Ga^{3+}$ species, uniformly dispersed throughout the channels of zeolite. The framework $Ga^{3+}$ and $Al^{3+}$ species contribute to Brønsted acid sites, while the extra-framework $Ga^{3+}$ species provide the Lewis acid sites. The catalyst that had a 0.46 ratio of Ga/(Al+Ga) exhibited the best performance, suggesting an optimal concentration and balance between the Brønsted and Lewis acid sites. Figure 10 illustrates the changes in product distribution concerning the Ga/(Al+Ga) ratio. It is evident that the aromatic yield rises with an escalation in the Ga/(Al+Ga) ratio, reaching a maximum of 55% at a ratio of 0.46. Conversely, the yield of $C_4$ (paraffins and olefins) decreases with an increase in this ratio, while no significant changes are observed in the cases of $C_1$, $C_2$, and $C_2^=$ with variations in the Ga/(Al+Ga) ratio.

### 2.2.2. Effect of Si/(Al+Ga) Ratio

In the second series of gallo-alumino-silicates (Cat-7 to Cat-10), the ratio of Si/(Al+Ga) was systematically adjusted from 11 to 65. Throughout this series, the ratio of Ga/(Al+Ga) was held constant to discern the direct impact of the Si/(Al+Ga) ratio on the aromatization of propane. This experimental design allows for the focused investigation of how changes

in the ratio of Si/(Al+Ga) specifically influence the process of propane aromatization. The data related to the catalytic evaluation of this series of catalysts are presented in Table 5 and Figures 11 and 12. A drastic drop in the propane conversion (80% to 18.3%) and aromatic yield (50% to 9.1%) was observed with the rise in Si/(Al+Ga) from 11 to 65. This behavior can be linked to the drop in acidity (NH$_3$-TPD) of catalysts with increases in the ratio of Si/(Al+Ga). As discussed in the above section, the aromatization of propane passes through several transitional reactions, e.g., cracking, oligomerization, dehydrogenation, cyclization, etc. All of these reactions depend upon the total acidity (Lewis and Bronsted) of the zeolites. A comparable trend has been observed by Phatansri et al. [38]. In this investigation, the ratio of Si/(Al+Ga) was modified by varying the amount of gallium, while maintaining a constant ratio of Si/Al. Their results indicated a rise in the conversion of C$_3$ and aromatic yield with a decrease in the ratio of Si/(Al+Ga). Figure 12 illustrates the alterations in product distribution concerning the Si/(Al+Ga) ratio. It is evident that the yield of almost all products, including aromatics, C$_3$=, C$_2$, and C$_1$, decreases with a rise in the ratio of Si/(Al+Ga). This behavior can be linked with changes in the acidity of the catalysts.

**Table 5.** Product distribution with respect to [Si/(Ga+Al)] ratio.

| Catalyst Code | Cat-7 | Cat-8 | Cat-9 | Cat-10 |
|:---:|:---:|:---:|:---:|:---:|
| Ratio Si/(Ga+Al) | 11.0 | 26.0 | 44.0 | 65.0 |
| C$_3$ Conversion | 80.0 | 60.0 | 36.0 | 18.3 |
| C$_1$ | 7.1 | 5.5 | 5.5 | 2.4 |
| C$_2$ | 7.5 | 6.5 | 5.2 | 2.3 |
| C$_2$= | 6.0 | 4.1 | 2.3 | 1.0 |
| C$_3$ | 20.0 | 40.0 | 64.0 | 81.7 |
| C$_3$= | 8.4 | 7.8 | 3.1 | 1.3 |
| C$_4$+C$_4$= | 1.0 | 1.1 | 2.0 | 2.2 |
| Total Aromatics | 50.0 | 35.0 | 18.0 | 9.1 |

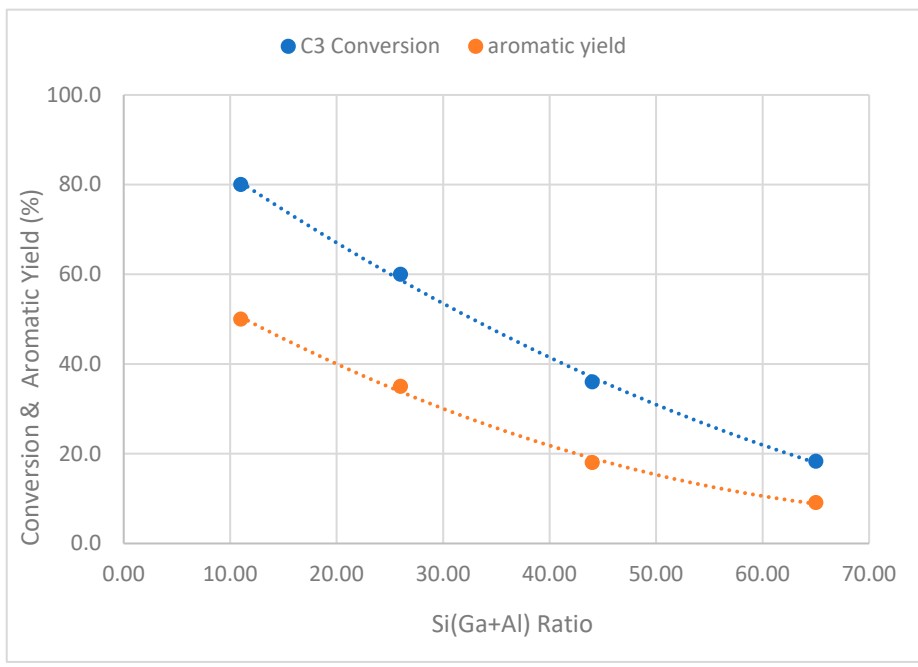

**Figure 11.** Plot of C3 conversion and aromatic yield against [Si/(Ga+Al)] ratio of catalyst samples.

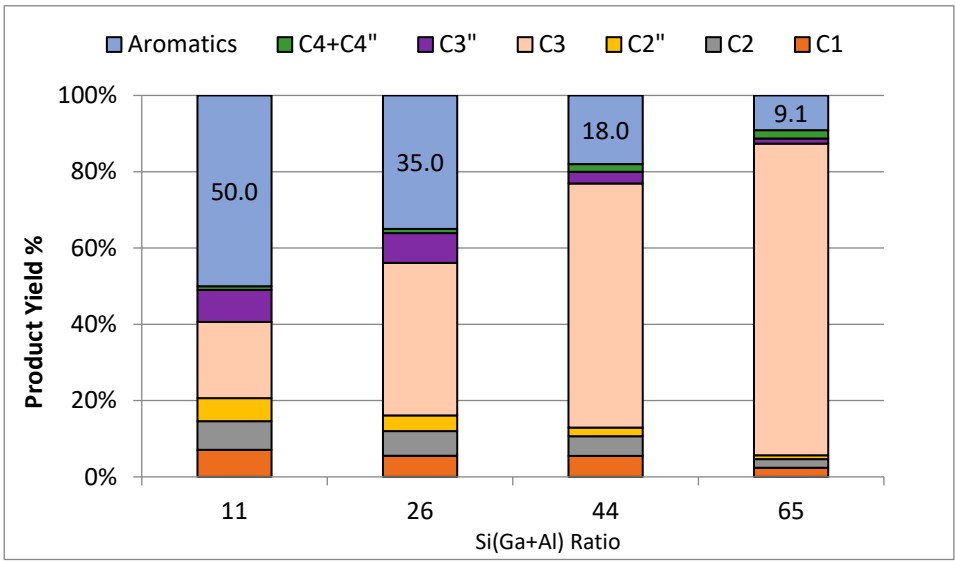

**Figure 12.** Distribution of product against [Si/(Ga+Al)] ratio of catalyst samples.

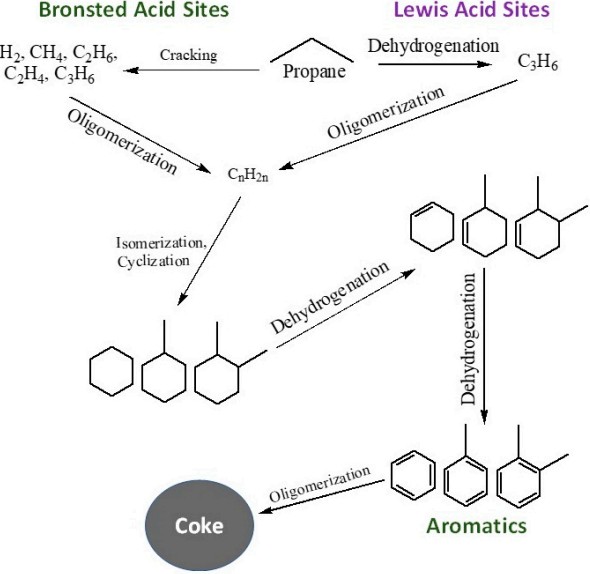

**Scheme 1.** A general reaction mechanism during aromatization of propane over Bronsted and Lewis acid sites.

## 3. Materials and Methods

### 3.1. Materials

Na$_2$SiO$_3$ solution, TPABr, Ga(NO$_3$)$_3$.nH$_2$O, Al$_2$(SO$_4$)$_3$ nH$_2$O, H$_2$SO$_4$, NaCl, and NH$_4$NO$_3$ were procured from Sigma Aldrich and were utilized without additional purification processes.

### 3.2. Catalyst Synthesis

Gallo-alumino-silicates were systematically synthesized through the hydrothermal crystallization process, involving the creation of a gel with components including sodium silicate, gallium nitrate, aluminum sulfate, tetrapropyl ammonium bromide (TPABr), sodium chloride, and sulfuric acid. The synthesis was performed at 180 °C for 72 h.

In a typical experiment, the zeolite synthesis was performed by the preparation of two distinct solutions. The first solution was obtained by diluting a required quantity of Na$_2$SiO$_3$ solution with distilled water. Concurrently, the second solution was prepared

by dissolving the required amounts of $Al_2(SO_4)_3$, $Ga(NO_3)_3$, TPABr, $H_2SO_4$, and NaCl in distilled water. Subsequently, both solutions were thoroughly mixed, stirred vigorously, and left to age overnight at room temperature. In the next step, this mixture was transferred into an autoclave reactor vessel and was subjected to the hydrothermal crystallization process at 180 °C with a rotation speed of 13–14 rpm for 72 hours. The hydrothermal crystallization process was stopped by cooling the autoclave reactor vessel to room temperature. The resulting zeolite was then filtered, washed extensively using distilled water, and dried at 120 °C for 3 h. Finally, the zeolite crystals were calcined at 550 °C using a heating rate of 5 °C/min for 3 h. These zeolites were further converted to acid-type by performing an ion-exchange reaction with ammonium nitrate solution. In this regard, one gram of zeolite was added to the ten milliliters of ammonium nitrate solution (1.0 M) and was stirred continuously for 60 min at 80 °C. The solution was then cooled down to room temperature, filtered, and washed with plenty of distilled water. This procedure was repeated three times and then solid material was dried in an oven at 100 °C for 12 h and then calcined at 550 °C for 3 h.

The details of zeolites prepared with varying elemental ratios are given in Table 1.

### 3.3. Catalyst Characterization

The gallo-alumino-silicate catalysts were subjected to the characterization through various instrumental techniques. The metal analysis of catalysts was performed using ICP-OES manufactured by Horiba.

The XRD patterns of catalyst samples were obtained using powder XRD by Rigaku. The XRD diffractions were recorded from 5 to 60° (2θ) with the detector speed of two degrees per minute and an increment of 0.02 degree per step. The relative crystallinity of gallo-alumino-silicate catalyst samples was determined using ASTM D5758 [40]. In this regard, five structural peaks in the 2θ region of 22.5–24.5° were integrated and then relative crystallinity was calculated using a commercial ZSM5 zeolite as a reference material using the following equation.

$$Relative\ Crystallinity\ (\%) = \frac{\sum area\ of\ \text{peaks for } catalyst \text{ in 2θ region of } 22.5 - 24.5°}{\sum area\ of\ \text{peaks for } reference\ ZSM5\ in \text{ 2θ region of } 22.5 - 24.5°} \times 100$$

The N2 desorption–adsorption isotherms were measured using Micromeritics equipment at −195 °C.

The samples were subjected to vacuum at 220 °C for 80 min. The acidity of the samples was assessed through $NH_3$-TPD experiments using BELCAT system from Japan. In the experiment, the samples underwent heating to 500 °C under helium for one hour, subsequently cooled to 100 °C. Afterward, the samples were subjected to ammonia at the same temperature (100 °C) for a duration of 30 min. Following that, the sample underwent a two hour helium flush to eliminate any excess $NH_3$. The furnace temperature was then gradually increased from 100 to 800 °C using helium, and the released ammonia was quantified using TCD.

The chemisorption-FTIR spectra of adsorbed pyridine were recorded using Nicolet-Is10 FTIR spectrometer to determine the Bronsted and Lewis acid sites. The catalyst samples were compressed into small pellets (0.12 g and 20 mm in diameter). The pellets were evacuated under a high vacuum at 450 °C for 4 h in a quartz cell. It was then contacted with pyridine vapors at 150 °C for 5 min. This was followed by evacuation at 150 °C for 1 h. The quartz cell was cooled down to room temperature and placed in an IR beam compartment and transmission spectra were recorded. The total number of Bronsted and Lewis acid sites were determined using equations proposed by Emeis [41].

The MAS-NMR measurements of $^{27}Al$, $^{29}Si$, and $^{71}Ga$ were obtained using a 400 MHz instrument manufactured by Bruker. The conditions for obtaining the NMR spectra were specified for each nucleus. The chemical shifts for $^{27}Al$, $^{29}Si$, and $^{71}Ga$ were calculated using $(NH_4)Al(SO_4)_2$, $C_6H_9D_6NaO_3SSi$, and $Ga(NO_3)_3$ reference samples. The surface

properties of the catalyst samples were analyzed by XPS using VG Escalab MkII photoelectron spectrometer.

The SEM analysis of catalyst samples was performed using high resolution desktop SEMoscope equipped with compact EDS detector manufactured by Inovenso. Fine powder of catalyst samples was spread on carbon tape and was quoted with gold to obtain high quality images.

### 3.4. Catalyst Evaluation

The aromatization of propane was conducted in a laboratory-scale fixed-bed reactor system as shown in Figure 13. The reactor tube (SS) was packed with 1.0 mL of sieved (500–1000 μm) catalyst particles in the middle along with the packing of inert silicon carbide particles above and below the catalyst. The catalyst was activated under flow of nitrogen (10 mL/min) for one hour at 550 °C. The reactor was then fed with nitrogen and propane in a molar ratio of 2:1. All reactions were executed at 550 °C, with GHSV of 1600 $h^{-1}$ and TOS of 5 h. The online GC equipped with two detectors, i.e., TCD and FID was utilized for characterization of products. The detailed hydrocarbon analysis was obtained using DHA software of Agilent. The flow diagram of fixed bed reactor is shown in Figure 11.

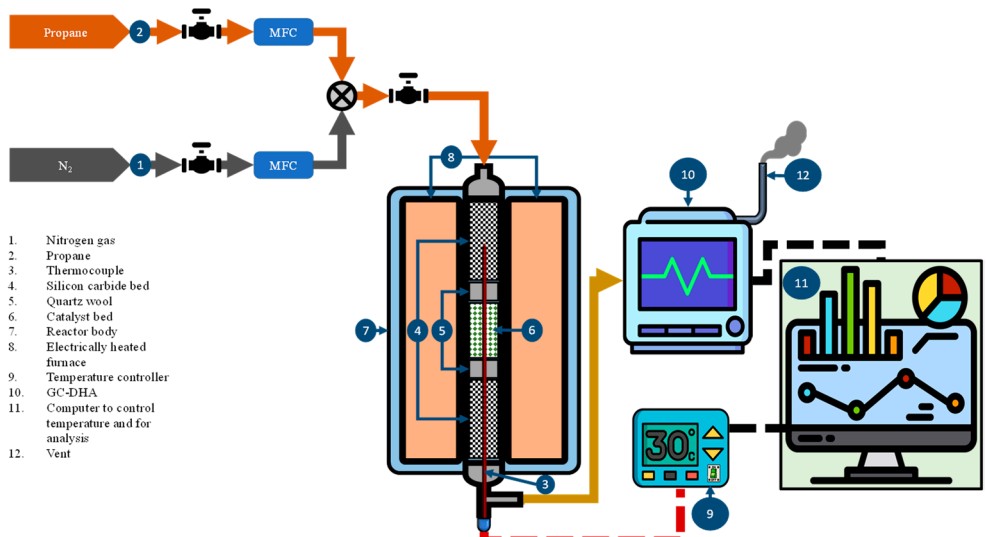

1. Nitrogen gas
2. Propane
3. Thermocouple
4. Silicon carbide bed
5. Quartz wool
6. Catalyst bed
7. Reactor body
8. Electrically heated furnace
9. Temperature controller
10. GC-DHA
11. Computer to control temperature and for analysis
12. Vent

**Figure 13.** Flow diagram of the fixed bed reactor.

### 4. Conclusions

This research has demonstrated that the ratios of Ga/(Al+Ga) and Si/(Al+Ga) in gallo-alumino-silicate catalysts have a significant impact on the yield of aromatics and propane conversion. The catalyst with optimal ratios of Ga/(Al+Ga) and Si/(Al+Ga) of 0.46 and 11.0, respectively, resulted in the highest conversion of propane (83.0%) and aromatic yield (55.0%). A slight decrease in the concentration of Bronsted acid sites and an increase in Lewis acid sites was observed with an increase in gallium content in the gallo-alumino-silcates. The total acidity (NH$_3$-TPD) dropped from 1.45 to 0.57 with a rise in the ratio of Si/(Al+Ga) from 11.0 to 65.0. The multinuclear MAS NMR study confirmed the isomorphous replacement of Al$^{3+}$ with Ga$^{3+}$ in the framework of gallo-alumino-silicates. The superior performance of gallo-alumino-silicate catalysts has been linked to the presence of dispersed extra-framework gallium species formed in the close proximity to the Bronsted acid sites.

**Supplementary Materials:** The following supporting information can be downloaded at: https://www.mdpi.com/article/10.3390/catal14030196/s1, Figure S1: XRD patterns of catalyst samples, Figure S2: NH$_3$-TPD profiles of catalyst samples, and Figure S3: FTIR spectra of chemisorbed pyridine of selected catalyst samples.

**Funding:** The author acknowledges the support provided by the Interdisciplinary Research Center for Refining and Advanced Chemicals (CRAC) at King Fahd University of Petroleum and Minerals (KFUPM) for funding the project # INRC2201. The support of KFUPM, Dhahran, Saudi Arabia, is also highly appreciated.

**Data Availability Statement:** The data presented in this study are available.

**Conflicts of Interest:** The author declares no conflicts of interest.

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
