# Peer review of "Transformation of Light Alkanes into High-Value Aromatics"

_catalysts, doi:10.3390/catal14030196_

Round 1
Reviewer 1 Report
Comments and Suggestions for Authors
In this manuscript, the author discussed the compositional effects on the conversion of light alkanes to aromatics using zeolite materials. They found that there is a strong dependence on the ratio of Ga and Si. However, some discussions in the manuscript are unclear and misleading, and is not suitable for publication in its current form.
1. 111-113. “a slight decrease in in the intensity of peak (50.2 ppm) related to AlTd species was observed with increase in Ga/(Al+Ga) ratio indicating a slight decrease in the framework Al species”. Decrease in framework Al should decrease both Td Al and Oh Al, why is only Td Al decreasing in NMR intensity?
2. 150. Why is the propane conversion defined this way? Why is there 100 at the front?
3. 187-188. How can Ga3+ and Al3+ be Bronsted acid sites? What are the pKas for these two sites? And what are the evidence that the authors have for this claim?
4. Table 1. I suggest add the absolute amount of Si, Al, and Ga for column 2-4. Also, the third column is essentially the ratio between the second and fourth. But for Cat-10, this does not work, why?
5. Figure 1 and 2. I suggest adding the original data for these two plots.
6. 312-313. What does the author mean by “extra-framework gallium species”?
Comments on the Quality of English LanguageN/A
Author Response
Thanks a lot for your comments.
I have replied to all of your comments point by point in the attached file.

Reviewer 2 Report
Comments and Suggestions for Authors
This paper is focused on the transformation of light alkane into high value aromatics using Ga/(Al+Ga) and Si/(Al+Ga) catalysts. The research content is interesting and meaningful, but there are still some issues that need to be corrected or improved. Some details need to be corrected:
1. The abstract should indicate which catalyst performs best, propane conversion and aromatic yield detailed data, rather than using words such as increasing, optimum value and declining.
2. There are many detail errors, such as -OSi, -O-Si, Figures 7, Figures-7, and Figure-6.
3. The structure-activity relationship of the catalyst should be explained clearly.
4. The format of the references should be revised.
5. The authors should describe the tests in a clear way to enhance the understanding of this work.
6. The conclusion is too simple. Some important conclusions should be included.
Comments on the Quality of English LanguageMinor editing of English language required.
Author Response
Thanks a lot for your comments.
I have given reply to all of your comments point by point in the attached file.

Reviewer 3 Report
Comments and Suggestions for Authors
In this work, Akhtar et al. mainly focused on the transformation of light alkane (propane) into high value aromatics using gallo-alumino-silicate catalysts. Propane conversion, aromatic yield, and aromatization/cracking ratio exhibited an increase with an increasing Ga/(Al+Ga) ratio, reaching an optimum value of 0.46 before declining. Conversely, An appreciable drop in conversion of propane and yield of aromatics was detected with the rise in Si/(Al+Ga) ratio, attributing to the decline in acidity. This manuscript is comprehensive but not innovative enough and lacks valid support for the experimental results:
1. “The XRD patterns of zeolites and their relative percent crystallinity are illustrated in Figure-S1 and Table 2, correspondingly.” Table 2 lists the relative crystallinity of zeolite in detail, but does not explain the calculation method of relative crystallinity, the basis for calculating relative crystallinity is not clear, please add details.
2. “There was no substantial difference in BET surface area and micropore volume with increase in Ga/(Al+Ga) ratio, indicating that addition of Ga3+species does not affect the microporous structure of catalyst samples. However, it was noticed that mesoporous surface area and mesopore volume were successively augmented with rise in Ga/(Al+Ga) ratio indicating an improvement in the diffusion of molecules of reactants and products.” The addition of Ga will increase the mesopore surface area and mesopore volume, which is conducive to the diffusion of reactants and product molecules, which can improve the efficiency of aromatization reaction. However, the manuscript does not explain in detail how the addition of Ga changes the mesopore volume.
3. On page 5, “Weak (Lewis) acid sites were increased with a rise in the ratio of Ga/(Al+Ga), while no significant change in strong acidity was noticed. The larger radius of Ga3+relative to Al3+results in a stress on the crystal lattice during isomorphous replacement, leading to an increase in the percentage of extra-framework Ga3+species” NH3-TPD can only analyze the acidity of zeolite, and cannot evaluate the acid properties of zeolite (Lewis). There are obvious errors in the manuscript. It is suggested to explain the acid properties of zeolite by adding py-IR.
4. The peaks of SI-O with different chemical shifts in FIG. 5 need to be fitted, and the percentage of Si in zeolite under different chemical environments is calculated using the peak area as an index.
5. The manuscript requires different SEM and TEM characterization to explain the morphology of zeolite, the distribution of Ga and the size of Ga grains in detail.
6. It is suggested to supplement the flow diagram of the fixed bed reactor.
7. Section 2.2 of the manuscript should explain the reaction path diagram of the aromatization of propane in detail, and analyze the modulation process of the reaction path by the active components of zeolite in depth, so as to correlate the structure of zeolite with the reaction results.
8. The English in the Introduction section of the manuscript needs to be strengthened, and the logic needs to be expressed more clearly.
Comments on the Quality of English Language
Minor editing of English language required
Author Response

(The authors gave the same response as above.)

Round 2
Reviewer 3 Report
Comments and Suggestions for Authors
It can be accepted now.
Author Response
The reviewer has written "It can be accepted now"
I have replied to the comments of academic editors in the cover letter